# A Combination of Resveratrol and Quercetin Prevents Sarcopenic Obesity: Its Role as a Signaling Inhibitor of Myostatin/ActRIIA and ActRIIB/Smad and as an Enhancer of Insulin Actions

**DOI:** 10.3390/ijms26104952

**Published:** 2025-05-21

**Authors:** Agustina Cano-Martínez, Jimena Alejandra Méndez-Castro, Viviana Estefanía García-Vázquez, Elizabeth Carreón-Torres, Eulises Díaz-Díaz, María Sánchez-Aguilar, Vicente Castrejón-Téllez, María Esther Rubio-Ruíz

**Affiliations:** 1Department of Physiology, National Institute of Cardiology Ignacio Chávez, Juan Badiano 1, Sección XVI, Tlalpan, Mexico City 14080, Mexico; agustina.cano@cardiologia.org.mx (A.C.-M.); mndzc.alejandra@gmail.com (J.A.M.-C.); vivianavaz1003@gmail.com (V.E.G.-V.); vicente.castrejon@cardiologia.org.mx (V.C.-T.); 2Department of Molecular Biology, National Institute of Cardiology Ignacio Chávez, Juan Badiano 1, Sección XVI, Tlalpan, Mexico City 14080, Mexico; juana.carreon@cardiologia.org.mx; 3Department of Reproductive Biology, National Institute of Medical Sciences and Nutrition Salvador Zubirán, Vasco de Quiroga 15, Sección XVI, Tlalpan, Mexico City 14000, Mexico; eulisesd@yahoo.com; 4Department of Pharmacology, National Institute of Cardiology Ignacio Chávez, Juan Badiano 1, Sección XVI, Tlalpan, Mexico City 14080, Mexico; maria.aguilar@cardiologia.org.mx

**Keywords:** sarcopenic obesity, myostatin, mass muscle, insulin, polyphenols, resveratrol, quercetin, metabolic syndrome

## Abstract

Sarcopenic obesity (SO), characterized by an excess of fat and a decrease in muscle strength or mass, is a global public health concern and is linked to metabolic conditions such as metabolic syndrome (MetS). Different mechanisms contribute to SO, such as inflammation, fatty acid infiltration, and insulin resistance (IR). Recently, myostatin (MYOST), an inhibitory factor for skeletal muscle tissue, was proposed as an aimed compound for the treatment of conditions of muscular metabolic imbalance mass and MetS. On the other hand, a therapy with natural compounds such as resveratrol (R) and quercetin (Q) is effective for the treatment of MetS, but its effect on the MYOST pathway has been poorly explored. The control group received water, and the MetS group received 30% commercial sugar in the drinking water for 6 months. Polyphenol mix (R at a dose of 50 mg/kg/day and Q at 0.95 mg/kg/day) was administered for 1 month. MetS rats present SO linked to an increase in the expression of MYOST/ActRIIA and ActRIIB (*p* < 0.0001). R+Q treatment prevented SO by lowering the expression of MYOST and its receptors and increased the expression of Smad 7 in MetS rats (*p* < 0.0001). Moreover, the polyphenol treatment reverted IR by increasing Akt phosphorylation, leading to an increase in muscle mass. It decreased lipid stores, restored glycogen accumulation, and increased myosin expression (*p* < 0.0001). The results of this work indicate that R+Q supplementation could be a promising therapeutic agent to prevent SO and sarcopenia derived from other metabolic alterations.

## 1. Introduction

Metabolic syndrome (MetS) is considered a chronic and progressive condition that includes a group of risk factors associated with other pathologies such as sarcopenia, which is defined as the loss of muscle mass and strength [1,2,3]. Recently, the combination of excess fat mass (obesity) and low skeletal muscle mass and function (sarcopenia) has been defined as sarcopenic obesity (SO), and it is being investigated due to its increase in prevalence worldwide [4]. There is communication between skeletal muscle and the normal and elevated adipose tissue present in MetS and other metabolic diseases, which is mediated by myokines and adipokines. This communication may lead to a negative feedback loop, which aggravates SO and insulin resistance (IR) [5].

Skeletal muscle is a very flexible and responsive tissue, able to undergo remodeling due to alterations. It constitutes around 40% of an adult’s total body mass and is responsible for 70–90% of insulin-induced glucose clearance. Insulin-stimulated skeletal muscle stores glucose mostly as glycogen. Myofibers contain structural proteins including actin, tropomyosin, troponin, and myosin, which bind to adenosine triphosphate (ATP), and depending on the composition of the myosin, different heavy chain (MYH) isoforms determine muscle fiber types [6]. Slow oxidative type 1 muscle fibers contain slow MYH 1 and fast glycolytic type 2 [7].

The exact etiology of SO remains unclear despite the suggestion of various mechanisms. There are different mechanisms that contribute to SO in association with MetS, such as persistent inflammation, lipotoxicity caused by accumulation of lipids (myosteatosis), oxidative stress, and IR. There are also alterations in growth hormone, testosterone, and estradiol that negatively affect standard of living and lead to greater frailty, weakness, dependency, morbidity, and mortality [8,9].

Myostatin (MYOST), referred to as growth and differentiation factor 8 (GDF 8), is part of the transforming growth factor-superfamily and functions as an inhibitor of skeletal muscle mass, influencing both the quantity and dimensions of muscle fibers. It binds to type IIB activin receptors (ActRIIB), which phosphorylates Smad 2 and Smad 3, resulting in the creation of a complex with Smad 4 [10,11,12,13]. This Smad heteromultimer acts as an inducible transcription factor that regulates skeletal muscle mass and is involved in muscle atrophy [10,11,14]. In addition, Smad 7 is a negative modulator of MYOST and transforming growth factor-beta (TGF-β) signaling by inhibiting Smad 2/Smad 3 phosphorylation, hence being an important mediator of muscle growth [15,16].

Many investigators have reported that MYOST levels are increased in diabetes, obesity, and muscular disease in rodent models. MYOST concentration is increased in patients with SO and is correlated with IR, according to Hittel (2009) [17]. A decrease in MYOST elevations results in a rise in muscle tissue through muscle hypertrophy and hyperplasia, therefore being considered as a novel therapeutic strategy still under investigation.

On the other hand, the phosphatidylinositol 3-kinase (PI3K)/Akt signaling pathway is essential for glucose absorption and lipid and protein metabolism and is important for regulating cell survival, growth, and apoptosis within skeletal muscle. In MetS, there is an occurrence of IR, and the protein synthesis pathway weakens, while degradation pathways become more active, resulting in decreased muscle mass. Moreover, there is cross-talk between the Akt signaling and MYOST pathways that down-regulates the Smad proteins [18,19,20].

Specific dietary supplementation with natural chemical compounds and physical training is crucial for modulating metabolic and molecular pathways related to the prevention and management of MetS and SO. Some evidence in clinical trials and animal models indicates that polyphenols like curcumin, quercetin, and resveratrol can inhibit muscle loss through their anti-inflammatory and antioxidant actions by reversing mitochondrial dysfunction and decreasing intramuscular fat accumulation [12,13,21,22,23]. As far as we know, this is the initial study that assesses the impact of the combination of resveratrol and quercetin (R+Q) on the signaling pathways that are involved in muscle growth.

This research was designed with the purpose of exploring whether supplementation with R and Q has beneficial effects on SO in rats with MetS. Modifications in the expression of MYOST/ActRIIA and ActRIIB/Smad signaling pathways were investigated as potential underlying regulatory mechanisms. Moreover, due to IR, a hallmark of MetS, which is also involved in the pathogenesis of SO, changes in the proportion of muscular fat deposition, glycogen deposits, and protein synthesis (myosin heavy chain, MYH1, expression) associated with the reestablishment of insulin sensitivity through the phosphorylation of Akt were also evaluated.

## 2. Results

Table 1 indicates that the test subjects exhibited MetS, marked by increased central fat accumulation, high blood pressure, dyslipidemia (raised triglycerides and non-HDL-C levels with decreased HDL-C levels), IR (HOMA-IR), and hyperinsulinemia. The R+Q treatment notably decreased levels of triglycerides and non-HDL-C, elevated HDL-C levels, and normalized the HOMA-IR in the MetS group. The administration of natural compounds did not significantly affect the parameters evaluated in the control group. Upon euthanasia, right soleus muscle wet weights were examined and adjusted according to the weight of the animals to assess changes in muscle mass. As expected, MetS rats developed SO due to the significantly lower muscle mass ratio when compared to control animals. The R+Q treatment enhanced the muscle mass in the MetS group, while there were no differences observed between control and C-R+Q groups (Table 1).

In SO, lipid infiltration is typically found in skeletal muscle; this study examines the presence of fat deposits in the soleus muscle by staining it with Oil Red. Results in Figure 1 show that MetS rats had an increase in lipid content in comparison to the control group muscles; however, the administration of R+Q to MetS animals induced a significative decrease (reaching up to 70%) in lipid content (Figure 1D), while in the control groups the lipid content remained unchanged.

To further investigate the mechanisms by which natural compounds prevent the reduction in muscle tissue, the expression of the MYOST signaling pathway was determined. There was a significant increase (*p* < 0.0001) in expression levels and co-localization of MYOST and its type IIA receptor in muscle sections of MetS rats; however, the administration of the mix R+Q was able to decrease their expression compared to the MetS group. The values in the expression level of MYOST and ActRIIA in the control groups did not show significant differences (Figure 2).

It has been reported that MYOST utilizes ActRIIB as a cooperating receptor to signal directly to muscle fibers; thus, we also evaluated the expression of this receptor in the samples from all experimental groups. The results in Figure 3 indicate that ActRIIB was present in greater quantities in soleus muscle from MetS rats than in muscle from control animals. The administration of polyphenols to MetS rats significantly decreases (*p* < 0.0001) the expression of this receptor (Figure 3).

Next, we analyzed if the consumption of R+Q was linked with changes in the expression of p-Smad 2/3, downstream components of the MYOST signal transduction pathway that are involved in sarcopenia forming a complex with Smad 4 by leading protein degradation. Our data in Figure 4 showed that the treatment with R+Q did not modify the protein expression levels of Smad 2/3, p-Smad 2/3, and Smad 4 in muscles from control and MetS animals. Interestingly, we observed that Smad7, which inhibits muscle atrophy by attenuating MYOST signaling, was significantly upregulated in soleus muscle from control and MetS rats treated with R+Q as compared to their respective group without treatment. The increase is more evident in MetS animals (31% vs. 117%, respectively) (Figure 4D).

Since IR is associated with poor muscle mass by mechanisms that involve suppression of PI3K/Akt signaling leading to protein degradation and impaired glycogen synthesis, we decided to evaluate the expression of Akt isoforms in muscles from all experimental groups. As anticipated, in basal conditions, MetS muscles exhibited IR, demonstrated by reduced expression of the p-Aktser473 in comparison to control muscle, while the R+Q treatment reestablished the insulin sensitivity in MetS rats (Figure 5A and 5B, respectively). Nonetheless, the total Akt expression remained unchanged in the muscles of both control and MetS rats.

We evaluated the glycogen levels in muscle fibers by Periodic Acid–Schiff (PAS) staining to further strengthen our data. Results presented in Figure 6 show a decrease in glycogen accumulation in muscles from MetS animals, which is improved with the treatment with R+Q, whereas it did not change in the control group.

Finally, considering that the insulin insensitivity present in MetS might enhance protein breakdown, we analyzed whether R+Q supplementation has any effect on the content of MYH1. The expression of MYH1 in the muscle of rats with MetS was 4 times less than what was observed in the control group (Figure 7A,B). R+Q treatment in MetS animals increased the MYH1 content by approximately 3 times without having any effect on the control group (Figure 7C,D).

## 3. Discussion

MetS comprises a group of risk factors that is linked to cardiovascular complications and SO, a condition defined as a loss of skeletal muscle tissue due to the excess of adipose tissue. The mechanisms that trigger SO are complex and multifactorial and include factors from outside, like diet and exercise levels. The internal factors have not yet been fully elucidated and include oxidative stress, an increase in leptin concentration, persistent low-level inflammation, mitochondrial impairment, lipotoxicity, and IR [1,24]. Strategies such as nutritional supplementation and exercise have been suggested to prevent SO; however, an effective treatment has not been established.

Much of the research on polyphenols’ health benefits has been done in animals or cell cultures using different doses. People consume polyphenols in small amounts in foods; hence, targeted dietary supplementation with natural chemical substances is advised and has been shown to have a notable and direct role in regulating metabolic and molecular pathways linked to the prevention and management of MetS and obesity. The quality and active components in supplements can differ significantly between manufacturers, complicating the determination of a standard dosage. R and Q are safe when taken in doses up to 1–3 g daily for up to 3–6 months; however, higher doses are more likely to cause stomach upset or headache. In some pathological conditions, caution is advised with the use of R and Q. For example, in people with bleeding disorders, those taking anticoagulant or antihypertensive medications, or those undergoing surgery [25,26].

Numerous reports have also shown the positive impacts of polyphenols in the management of SO; nevertheless, as far as we know, there are no studies that show the impact of polyphenols on the MYOST signaling pathway analyzed in this paper. In this study, we demonstrate that the administration of a combination of R+Q was able to prevent SO by decreasing the expression of MYOST and its receptors (ActRIIA and ActRIIB) and by increasing the expression of Smad 7 in MetS rats. Moreover, the polyphenol treatment reverses local IR by increasing Akt phosphorylation, resulting in a rise in muscle mass, a reduction in lipid stores, restored glycogen accumulation, and increased myosin expression.

Our findings indicate that the MetS rats present an increase in central adiposity, hypertension, dyslipidemia, hyperinsulinemia, and IR. As expected, these male Wistar rats present SO due to the increase in abdominal fat, which was accompanied by a decrease in the muscle mass ratio compared to control rats (0.32 ± 0.03 vs. 0.51 ± 0.05). These findings support a number of studies that have documented the loss of muscle mass in animal models of obesity and MetS based on age and high-fat diet consumption (male Sprague-Dawley rats) [8,27]. Like previously reported, the administration of R+Q improved the parameters of our MetS model, which is easy to care for, relatively low cost, and emulates specific lifestyle factors that humans exhibit in the pathophysiology of MetS [28,29,30]. Moreover, R+Q treatment was able to prevent SO in MetS rats. Our findings are consistent with earlier research that demonstrated the connection of polyphenol supplementation with the rise in skeletal muscle mass in both clinical trials and animal models [13,20,31].

MetS and SO are accompanied by lipid deposition in muscle; this accumulation of intramuscular fat leads to diminished muscle contractile function and metabolic dysfunctions [32,33]. Our findings show that supplementation with R+Q considerably decreases lipid accumulation in skeletal muscle in MetS rats without having a significant effect in control animals. This beneficial effect of polyphenols on the decrease in muscular lipid deposition had been reported in human and animal studies (C57BL/6 mice and obese Wistar rats fed a high-fat diet) [34,35].

Although various mechanisms are suggested, the exact etiology of sarcopenia remains unclear. Many investigators have demonstrated the association of sarcopenia with the increase in MYOST, a muscle growth inhibitor that exerts its effects by binding to activin receptors, most notably activin receptor IIB [10]. We demonstrated that there was a significant increase in the expression levels of MYOST and its type IIA receptor in muscle sections of MetS rats. Recent research has also shown that natural substances like isoflavones, gingerol, curcumin, R and Q could be applied to prevent muscle atrophy because they inhibit the expression of MYOST [19,36,37]. However, as far as we know, this is the first report of a colocalization and decrease in the expression of both MYOST and ActRIIA in skeletal muscle of a MetS rat model by the administration of R+Q.

MYOST also restrains the growth of skeletal muscle through repression of proliferation, differentiation, and the synthesis of protein by binding to its ActRIIB receptor and the subsequent phosphorylation of Smad 2 and Smad 3. Our data indicate that ActRIIB was available in larger amounts in the soleus from MetS rats rather than muscle from control animals. The administration of polyphenols to MetS rats significantly decreases the expression of this receptor. Some authors previously reported that ActRIIB expression is increased in muscles from leptin-deficient ob/ob obese mice and streptozotocin-induced diabetic rats, which also occurred in our MetS model [38,39]. However, to the best of our knowledge, our data offer the initial proof that the R+Q supplementation decreases the expression of ActRIIA and ActRIIB, preventing SO. Resveratrol has mimetic properties to exercise [40] and physical training, and it decreases ActRIIB muscle expression through activating AMP-activated protein kinase (AMPK), a strong regulator of metabolism and gene expression in skeletal muscle [41]. Therefore, further research is necessary to evaluate the role of AMPK in our MetS model supplemented with R+Q to prevent SO.

Smad proteins are components of the MYOST and TGF-β signaling pathways, which regulate skeletal muscle mass, fibrosis, and tissue regeneration by controlling the transcription of genes that affect protein synthesis and degradation. Therefore, we decided to evaluate their expression. Our results showed that the treatment with R+Q did not modify the protein expression levels of Smad 2/3, p-Smad 2/3, and Smad 4 in muscles from control and MetS animals. Interestingly, we observed that Smad 7, which prevents Smad 2/3 phosphorylation and promotes the formation of the ActRIIB complex, was significantly upregulated in soleus muscle from control and MetS rats treated with R+Q as compared to groups without treatment. The increase is more evident in MetS animals (31% vs. 117%, respectively). Previous studies have shown that R treatment can modulate the Smad 7 expression as a form of intracellular negative feedback to prevent pathological processes in a variety of cells and tissues, such as skeletal and cardiac muscles, liver, and scar fibroblasts [42,43,44].

On the other hand, a hallmark of MetS is IR, which also plays an important role in sarcopenia. The PI3K/Akt signaling pathway plays a crucial role by promoting protein synthesis (via mTOR), blocking protein degradation, activating glycogen synthase kinase 3 (GSK3), and down-regulating the Smad pathways [19,20,45]. Our findings indicated that the muscles of MetS rats had a decrease in the expression of p-Aktser473 compared to control muscles. Muscle controls glucose homeostasis and accounts for 80% of the blood’s postprandial glucose absorption. Reduced insulin sensitivity in muscle is indicated by a lower uptake of glucose, as well as changes within the cells that affect glucose transport, phosphorylation, oxidation, and the synthesis of glycogen. In MetS rats that present compensatory hyperinsulinemia due to IR, glycogenesis is suppressed; these findings are similar to those reported for skeletal muscle of C57BL6 mice fed a high-fat diet, which exhibited a decline in the glycogen levels [46]. In our study, the soleus muscle of rats with MetS treated with R+Q presented glycogen levels equivalent to the control group. Our findings align with those noted in earlier research regarding the increase in insulin sensitivity in muscle by Q or R administration [47,48,49].

The content of structural proteins, including actin, tropomyosin, troponin, and myosin, which binds to ATP, and the composition of types of muscle fibers are determined by MHC isoforms [6]. The soleus muscle used in this study is rich in oxidative myofibers (slow) but also has a low content of fast myofibers, which have a glycolytic metabolism. Slow myofibers are characterized by a high content of MYH1, while fast myofibers have a high content of type II [50]. Although still controversial, a growing consensus suggests that natural compounds have a beneficial effect on myofiber remodeling and that R, curcumin, and Q can modulate muscle fiber-type size and composition through several signaling pathways [44,50,51,52]. Moreover, Pellegrinelli et al. (2015) [53] reported that the presence of contractile muscle proteins is diminished in obese patients. In the present study, we found that MYH1 levels in the soleus muscle of the MetS group are four times lower than in the control group. The R+Q supplementation restored the insulin sensitivity, which may also be involved in the increase in MYH1 expression. This effect on MYH1 content in myofibers may be similar to that reported by Nagai (2024) [50] in human skeletal muscle satellite cells treated with quercetin.

## 4. Materials and Methods

### 4.1. Animals and Treatment

The Wistar male rats employed in this study were provided by the Bioterio of our Institution. The research received approval from the animal ethics board of our institution (protocol #22-1342). Wistar male rats weighing 56 ± 3 g (21 days old) were used. Twelve rats were chosen at random for the control group and twelve for the MetS rats. Control rats received tap water for drinking, and group 2, MetS rats, received 30% sugar in their drinking water for 5 months. Fifty percent of each rat group was given a combination of R+Q every day for four weeks at doses of 50 mg/kg/day and 0.95 mg/kg/day, respectively (provided by ResVitalé TM, General Nutrition Centers (GNC), Pittsburgh PA, USA which contains 20 mg of Q per 1050 mg of R). Groups without R+Q treatment only received the vehicle. The mixture of R+Q had been previously dissolved in 1 mL ethanol solution (20%).

Groups of two rats were housed in polycarbonate (480 mm × 375 mm × 210 mm, Eurostandard type IV) ventilated cages (3 cages per group) within a ventilated cage rack (from Tecniplas, Buguggiate, Italy) in a ventilated room at 18 air changes hourly. The cages contained autoclaved wood chip bedding material (from Bioinvert, Estado de México, México) and were equipped with a polycarbonate tunnel for enrichment. The diet comprised a regular chow diet (LabDiet 5001, PMI Nutrition International, LLC., Brentwood, MO, USA) ad libitum. A regulated temperature and a 12:12 h light/dark cycle were controlled in the room where the animals were kept. Systolic arterial blood pressure was assessed in awake animals employing the tail cuff technique as outlined before [28,54,55].

### 4.2. Serum Biochemical Assessment

After overnight fasting, the rats were humanely euthanized without anesthesia. After collection of the whole blood in a centrifuge tube (without anticoagulant), the blood was allowed to clot at room temperature for 15 min and then centrifuged at 2500 rpm for 10 min to separate the clot. Following centrifugation, the serum was transferred into a microcentrifuge tube and was frozen for later biochemical analysis. Serum glucose, cholesterol, and triglycerides were measured by enzymatic colorimetric methods using reagents from Pointe Scientific (Pointe Scientific Inc., Canton Township, MI, USA) and an automated biochemical analyzer, Mindray BS-200 (Mindray Medical International Limited, Shenzhen, China). Serum concentrations of insulin were measured using RadioImmunoAssays (RIA) developed and validated in the “Salvador Zubirán” National Institute of Medical Sciences and Nutrition; insulin concentration was measured using a gamma counter Cobra II (Packard Instrument Company, Inc., Meriden, CT, USA) as has been previously described [56]. Through the homeostasis model [(HOMA-IR), (insulin (μU/mL) glucose (mmol/L)/22.5)], using fasting insulin and glucose levels, the IR was estimated [29]. After the ultracentrifugation of the plasma at a density of 1.063 g/mL for 2.5 h at 100,000 rpm (Beckman optima TLX, PAS, Brea, CA, USA), the high-density lipoprotein (HDL) cholesterol content was determined in the bottom fraction obtained [57]. The difference between the values of total cholesterol and HDL-C corresponds to the non-HDL-C [low-density lipoprotein cholesterol (LDL-C), intermediate-density lipoproteins (IDL), and very-low-density lipoproteins (VLDL)].

### 4.3. Tissue Sampling

The intra-abdominal white adipose tissue was carefully dissected with scissors after euthanasia, wet weight was determined, and then the tissue was discarded.

Immediately after rat euthanasia and hind limb trichotomy, soleus muscles were dissected through a skin incision at the level of tibialis anterior muscles. The weight of every muscle was adjusted to the body weight of the animals. Right soleus muscles were rapidly frozen in liquid nitrogen and kept at −70 °C until the analysis was performed. Each left muscle was covered with Allprotect Tissue Reagent (Qiagen, Hilden, Germany), spread on a solid OCT [Tissue-Tek (optimum cutting temperature)]-coated cork (previously kept at −20 °C), frozen in liquid nitrogen, and stored at −70 °C in a lidded container until processed for frozen sectioning.

### 4.4. Western Blot Analysis

Muscle was placed in a porcelain mortar, chilled in liquid nitrogen, and ground; powdered tissue was placed in a tube with a lysis buffer (25% *w*/*v*) (25 mM Hepes, 100 mM NaCl, 15 mM Imidazole, 10% glycerol, 1% Triton X-100, pH = 8) and Halt protease inhibitor cocktail-EDTA free (from ThermoScientific, Rockford, IL, USA) and then homogenized with a Teflon pestle. The homogenate was centrifuged at 19,954× *g* for 15 min at 4 °C; the supernatant was collected and stored at −70 °C. The protein concentration of each sample was assessed using the Bradford method [58]. Fifty micrograms of protein was isolated via SDS-PAGE and transferred to a PVDF membrane (Millipore Corp., Burlington, MA, USA). Using the following primary antibodies: anti-Smad 2/3 (ab63672 Abcam PLC, Cambridge, UK); anti-phospho-Smad 2/3 (ab63399 Abcam PLC, Cambridge, UK); anti-Smad 4 (ab40759 Abcam PLC, Cambridge, UK); anti-Smad 7 (ab21642 Abcam PLC, Cambridge, UK 8); anti-Akt pan (from Cell Signaling Technology, Danvers, MA, USA Cat# 2920); and anti-Phospho-Akt1 (Ser473) (from Cell Signaling Technology Danvers, MA, USA Cat# 9018) at a final dilution of 1:1000, the proteins were identified. Subsequently, the membranes were incubated overnight at 4 °C with a secondary antibody that is conjugated with horseradish peroxidase at a dilution of 1:10,000 (Santa Cruz Biotechnology, Santa Cruz, CA, USA). Every blot was treated with GAPDH antibody (sc-365062 from Santa Cruz Biotechnology, Santa Cruz, CA, USA) as a control. The signal was developed with chemiluminescence Immobilon Western Substrate (Millipore Corporation, MA, USA) using ChemiDoc™ workflow version 3 Touch Imaging System (Bio-Rad Laboratories, Hercules, CA, USA), and images were densitometrically quantified utilizing specific software (Image Lab™ version 6.1 Software, Bio-Rad, Hercules, CA, USA).

### 4.5. Histological Analysis

Prior to performing the histological sections, the tissues were cryopreserved with 30% sucrose and prepared with OCT. Transverse sections were made by freezing (10 µm) of the central part of each soleus muscle, in a cryostat (Triangle Biomedical Sciences (TBS) minotome plus cryostat). The sections were mounted on adhesion slides (StatLab), [each with 8 sections (from 4 different animals of each study group, two blocks)], which were stored at 4 °C until used in histochemical staining [(Oil Red—a fat-soluble dye utilized to give neutral lipids and triglycerides a deep red color—and Periodic Acid–Schiff (PAS) to detect polysaccharides such as glycogen)] and for immunolocalization by immunofluorescence assays (MYOST, ActRIIA, ActRIIB and MYH1). Five sets of slides with the sections of the soleus muscle were selected and dried (RT, 30 min).

#### 4.5.1. Oil Red Staining

The set of slides for lipid detection were fixed with buffered formalin and calcium (2% calcium chloride) for 1 h, followed by 3 washes (5 min each) with deionized water. The slides were incubated with Oil Red (saturated solution in isopropanol) for 30 min and washed 3 times (30 s each) with deionized water. The coverslip was placed with fluorescence mounting medium [Antifade Mounting Medium (Vethashielsd)] for later microscopic examination.

#### 4.5.2. Periodic Acid–Schiff (PAS) Staining

Sections for detecting glycogen were rehydrated in distilled water, fixed (4% PFA, 24 h), and subjected to PAS staining (HYCEL, Zapopan, JAL, MX, Cat-64695), with consecutive incubation: 0.5% PAS reagent, following the supplier’s instructions, washing with distilled water, and differentiated with hydrochloric acid. After dehydrating and clearing the histological sections, the coverslips were placed with entellan (mounting medium), and they were stored at room temperature for later microscopic analysis.

#### 4.5.3. Immunolocalization: Immunofluorescence Assay

A hydrophobic barrier [(ImmEdge^®^ Hydrophobic Barrier PAP Pen (H-4000)] was placed on the dried slides around the histological sections, rehydrating them with PBS. For the immunolocalization of MYOST, ActRIIA, and ActRIIB, prior to incubation with normal goat serum (5%), it was fixed with 4% PFA (10 min), and the autofluorescence was blocked with ammonium chloride (50 mM) [59].

After blocking nonspecific binding [(5% normal goat serum (1 h)], the slides were incubated with the corresponding primary antibody MYOST [(1/100) (10 µg/mL) (ab203076 Abcam PLC, Cambridge, UK)], ActRIIA [(1/50) (0.66 µg/mL) (ab78412 Abcam PLC, Cambridge, UK)], or ACTRIIB [1/100) (10 µg/mL) (bs-12417R from Bioss, Inc., Woburn, MA, USA) in a humid chamber at 4 °C overnight. Detection of MYOST and ActRIIA was performed simultaneously on the same slides. After incubation with the primary antibodies, the slides were incubated with the secondary antibodies: goat anti-rabbit [(1/1000) (AF488 –ab150077 Abcam PLC, Cambridge, UK)] and goat anti-mouse [(1/1000) (AF647) (ab15015 Abcam PLC, Cambridge, UK)] by incubation with MYOST and ActRIIA primary antibodies, respectively (60 min at RT). The goat anti-rabbit [(1/1000) (AF488 –ab150077 Abcam PLC, Cambridge, UK) was used for 60 min at RT for the detection of ActRIIB. After washing, the 4′,6-diamidino-2-phenylindole dihydrochloride (DAPI) was employed to visualize the nuclei, and finally the coverslip was placed with fluorescence mounting medium [Antifade Mounting Medium (Vethashielsd)] for later microscopic examination and image acquisition. For immunolocalization of MYH1, the slides were treated with 10% normal goat serum in PBS and incubated with the primary antibody anti-MYH1 [(1/150) (6.66 µg/mL) (ab91506 Abcam PLC, Cambridge, UK)] overnight at 4 °C in a humid chamber. After removing the primary antibody and washing, the slides were incubated with the secondary antibody anti-rabbit made in goat [(1/1000) (AF488 –ab150077 Abcam PLC, Cambridge, UK) for 60 min at RT. The slides were washed, and DAPI was used to visualize the nuclei, and finally the coverslip was placed with fluorescence mounting medium [Antifade Mounting Medium (Vethashielsd)] for later microscopic analysis.

#### 4.5.4. Image Acquisition and Label Quantification

Oil Red, MYOST, ActRIIA, ActRIIB, and MYH1 images were captured on a Floid Cell Imaging Station (Life Technologies, Carlsbad, CA, USA), and glycogen images were acquired with the Q-IMAGING camera, MicroPublisher 5.0 (Burnaby, BC, Canada), coupled to an Olympus BX51 (Hachioji, Japan), microscope. In each case, 4 (20×) fields were taken from each section (1 section from each animal (*n* = 4), by duplicate in each slide), with 32 fields in total per slide.

Quantification of fluorescence intensity per unit area, as integrated optical density [IOD (lum/pix^2)] of Oil Red, MYOST, ActRIIA, and ActRIIB, was performed using Image-Pro Premier version 9 software (MediaCybernetics) interfaced with a camera (Q IMAGING, MicroPublisher 5.0) integrated into an Olympus BX51 microscope. In the case of glycogen, the values correspond to the % of PAS label. On the images acquired for MYH1, the total number of myofibers, as well as the number of MYH1-positive myofibers, was manually quantified, and the ratio between the number of MYH1 myofibers/total number of myofibers was calculated.

### 4.6. Statistical Analysis

Results are expressed as the mean ± standard error of the mean (SEM). All the graphs and statistical analyses were performed using GraphPad Prism version 8.0 (GraphPad Software, Dotmatics, Boston, MA, USA). Student’s *t* test was used for comparisons between 2 groups. For multiple comparisons, we applied one-way analysis of variance (ANOVA) followed by a post hoc test (Tukey). Differences were considered significant when the *p* value was <0.05.

## 5. Conclusions

SO is a condition associated to MetS and it is linked to the increase in MYOST/ActRIIA and ActRIIB expression and with IR. The results in this work, using a MetS rat model with SO, provide a novel regulatory mechanism of the combination of R+Q to prevent SO by decreasing the expression of MYOST and its activin receptors and by increasing the expression of Smad 7. Moreover, the polyphenol treatment reverses IR by increasing Akt phosphorylation, leading to an increase in muscle mass, a decrease in lipid stores, and the restoration of glycogen accumulation and increased myosin expression. Therefore, this work is evidence that supplementation with R+Q could be a promising therapeutic measure to prevent SO, as well as sarcopenia derived from other metabolic alterations.

## Figures and Tables

**Figure 1 ijms-26-04952-f001:**
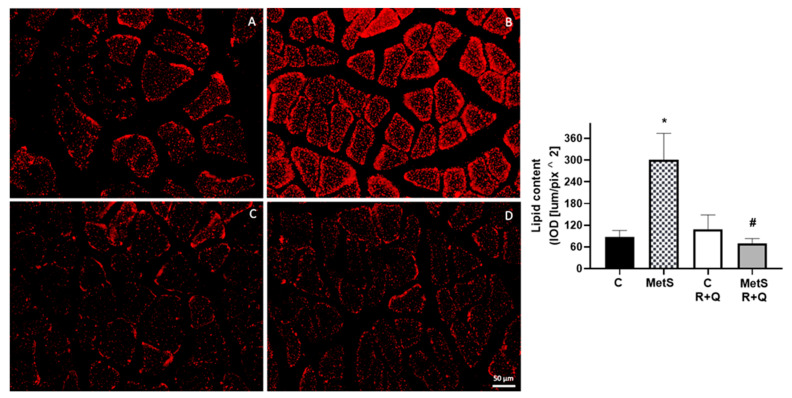
Resveratrol plus quercetin (R+Q) administration decreased lipid deposition inside skeletal muscle from MetS rats. Representative images stained with Oil Red are presented and the mean ± SEM of immunodetection levels is showed in the graph. * *p* < 0.0001 vs. control; # *p* < 0.0001 vs. MetS group. Panel (**A**) = control, panel (**B**) = metabolic syndrome (MetS), panel (**C**) = control plus R+Q, panel (**D**) = MetS plus R+Q. In each case, 4 fields (20×) were taken from each section (1 section from each animal (*n* = 4), by duplicate in each slide), with 32 fields in total per slide (*n* = 32/group).

**Figure 2 ijms-26-04952-f002:**
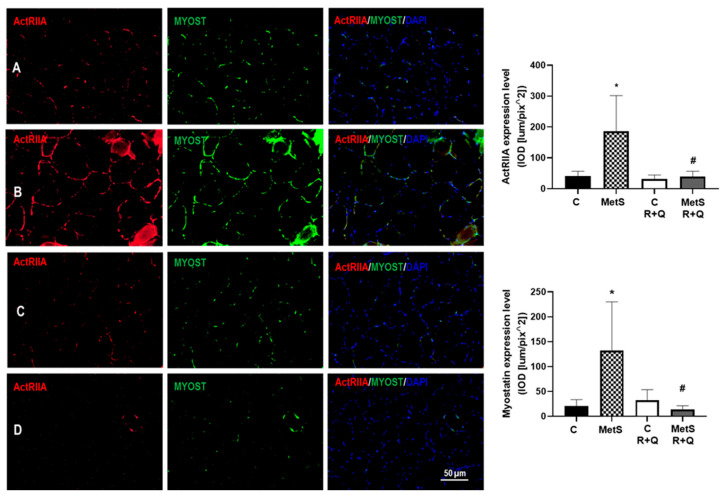
Resveratrol and quercetin (R+Q) treatment decreased the expression of MYOST and its ActRIIA receptor in muscle from MetS rats. The representative images illustrate the immunodetection of ActRIIA (red), MYOST (green), and 2-[4-(Aminoiminomethyl) phenyl]-1H-Indole-6-carboximidamide hydrochloride. DAPI was employed to identify the nuclei. The graph shows the mean ± SEM of the expression levels. * *p *< 0.0001 vs. control; # *p *< 0.0001 vs. MetS group. Panel (**A**) = control, panel (**B**) = metabolic syndrome (MetS), panel (**C**) = control plus R+Q, panel (**D**) = MetS plus R+Q. In each case, 4 fields (20×) were taken from each section (1 section from each animal (*n* = 4), by duplicate in each slide), with 32 fields in total per slide (*n* = 32/group).

**Figure 3 ijms-26-04952-f003:**
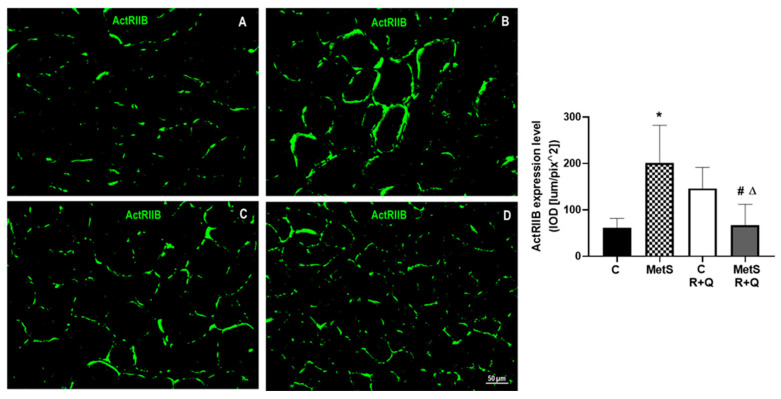
Resveratrol and quercetin (R+Q) treatment decrease the ActRIIB receptor expression in soleus muscle from MetS rats. Representative images of immunodetection are presented and the mean ± SEM of the expression levels are shown in the graph. * *p* < 0.0001 vs. control; # *p* < 0.0001 vs. MetS group; Δ *p* = 0.0002 vs. control plus R+Q. Panel (**A**) = control, panel (**B**) = metabolic syndrome (MetS), panel (**C**) = control plus RSV+QRC, panel (**D**) = MetS plus R+Q.

**Figure 4 ijms-26-04952-f004:**
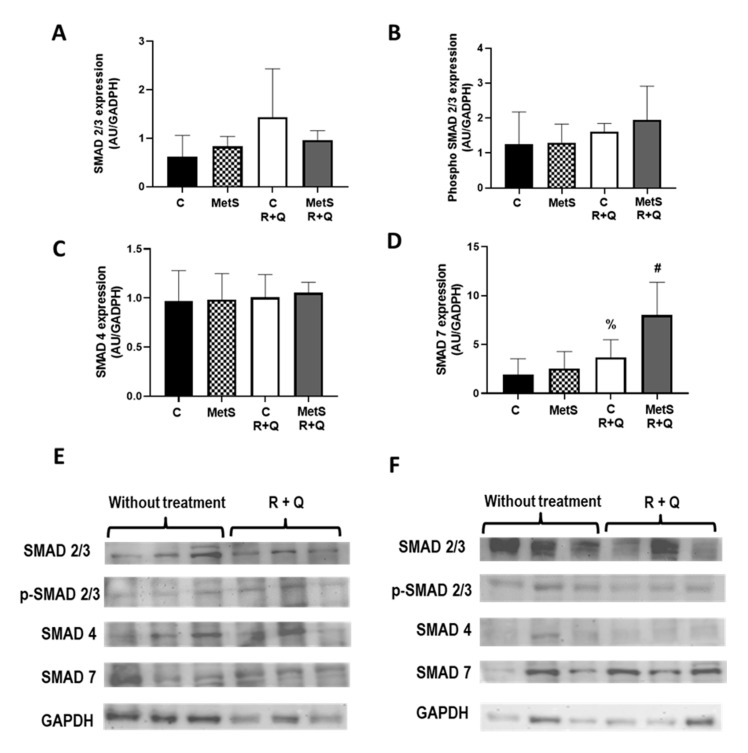
Effect of resveratrol and quercetin (R+Q) treatment on expression of muscle Smad signaling pathway. Expression was evaluated by Western blot in the soleus muscle from all experimental groups. (**A**) Smad 2/3; (**B**) p-Smad 2/3; (**C**) Smad 4; (**D**) Smad 7. Representative Western blot from control (**E**) and metabolic syndrome (MetS) rats (**F**). Data represent mean ± SEM are shown in the graphs (*n* = 5 per group). % *p* = 0.0006 vs. control group; # *p* < 0.0001 vs. MetS group.

**Figure 5 ijms-26-04952-f005:**
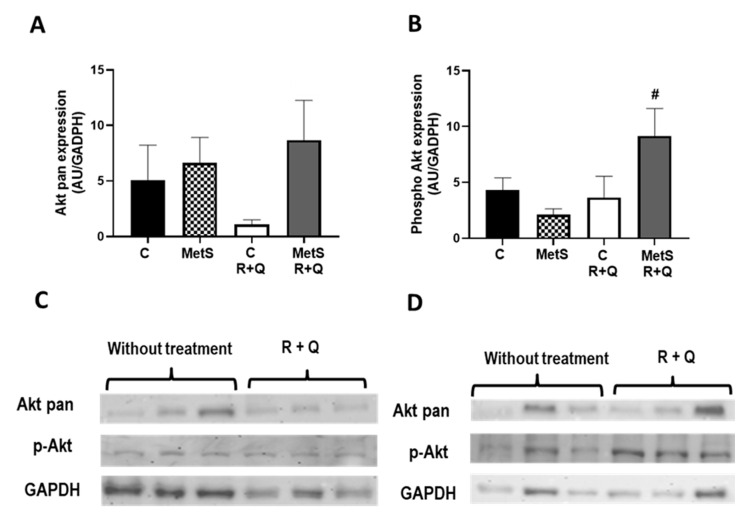
Resveratrol and quercetin (R+Q) supplementation improve muscle IR in MetS rats. Western blot analysis was employed to evaluate the expression in the soleus muscle from all experimental groups. (**A**) Akt pan; (**B**) p-Aktser473; (**C**) representative immunoblot from control group; (**D**) representative immunoblot from MetS group. Data represent mean ± SEM (*n* = 5 per group). # *p* < 0.0001 vs. MetS group.

**Figure 6 ijms-26-04952-f006:**
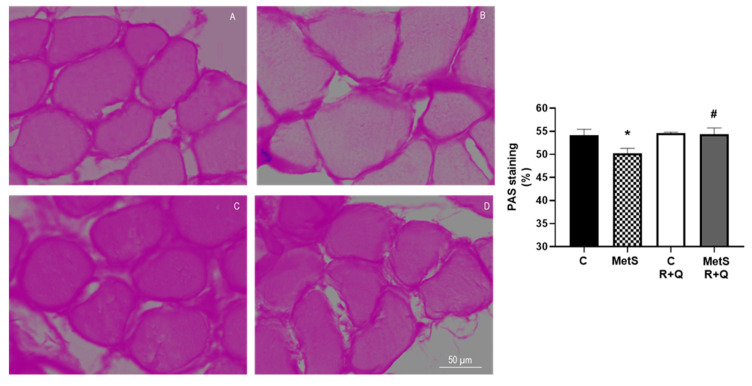
Representative pictures of PAS-stained sections of skeletal muscle from control and MetS rats treated with resveratrol plus quercetin (R+Q). Panel (**A**) = control, panel (**B**) = metabolic syndrome (MetS), panel (**C**) = control plus R+Q, panel (**D**) = MetS plus R+Q. The graph shows the mean ± SEM of the proportion of PAS-positive label. * *p* < 0.0001 vs. control; # *p* < 0.0001 vs. MetS group.

**Figure 7 ijms-26-04952-f007:**
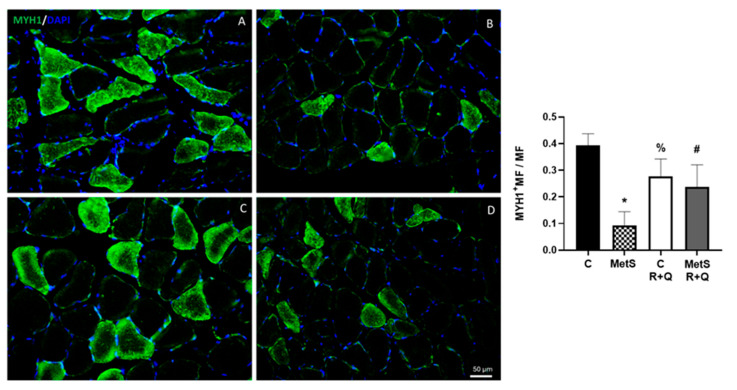
Resveratrol and quercetin administration increased the expression of myosin heavy chain in muscle from MetS rats. The representative images show the immunodetection of myosin heavy chain (MYH1) (green) and the nuclei were marked with DAPI. The graph shows the mean ± SEM of the number of MYH1-positive myofibers/total number of myofibers (MYH1+MF/MF). * *p* < 0.0001 vs. control; # *p* < 0.0001 vs. MetS group; % *p* = 0.0006 vs. control group. Panel (**A**) = control, panel (**B**) = metabolic syndrome (MetS), panel (**C**) = control plus R+Q, panel (**D**) = MetS plus R+Q.

**Table 1 ijms-26-04952-t001:** Body characteristics and biochemical parameters of control and metabolic syndrome (MetS) male (age 6 months) rats treated with R+Q.

	Control	Control Plus R+Q	MetS	MetS Plus R+Q
Weight (g)	457.2 ± 13.7	514.2 ± 20.9	598.1 ± 9.1 *	481.8 ± 7.7 #
Abdominal fat (g)	4.9 ± 0.3	6.3 ± 0.9	12.9 ± 0.3 *	7.8 ± 1.0 #
Blood pressure (mm Hg)	102.8 ± 0.8	110.8 ± 3.9	141.8 ± 0.9 *	115.4 ± 2.9 #
Total cholesterol (mg/dL)	53.7 ± 2.9	60.3 ± 3.5	63.5 ± 3.2	61.3 ± 1.5
HDL-c (mg/dL)	29.2 ± 1.9	28.4 ± 1.3	15.8 ± 1.2 *	22.3 ± 1.9 #
Non-HDL-c (mg/dL)	24.2 ± 1.4	20.5 ± 0.8	37.1 ± 4.2 *	20.1 ± 1.9 #
Triglycerides (mg/dL)	83.6 ± 6.7	78.9 ± 4.2	145.2 ± 6.2 *	98.3 ± 5.2 #
Glucose (mg/dL)	93.0 ± 3.3	91.1 ± 5.2	95.5 ± 2.4	90.1 ± 2.2
Insulin (μU/mL)	0.14 ± 0.03	0.12 ± 0.02	0.45 ± 0.05 *	0.16 ± 0.01 #
HOMA-IR	0.81 ± 0.19	0.56 ± 0.10	2.16 ± 0.30 *	0.60 ± 0.07 #
Muscle-to-body mass ratio (mg/g)	0.51 ± 0.05	0.49 ± 0.02	0.32 ± 0.03 *	0.44 ± 0.01 #

Values are mean ± SEM. Abbreviations: HOMA-IR = homeostatic model assessment of insulin resistance; HDL-c: high-density lipoprotein cholesterol. *n* = 6 animals per group; * *p* < 0.001 MetS vs. control; # *p* < 0.001 MetS plus R+Q vs. MetS group.

## Data Availability

Data are contained within the article.

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
