# Peer review of "A Combination of Resveratrol and Quercetin Prevents Sarcopenic Obesity: Its Role as a Signaling Inhibitor of Myostatin/ActRIIA and ActRIIB/Smad and as an Enhancer of Insulin Actions"

_ijms, 2025, doi:10.3390/ijms26104952_

Round 1
Reviewer 1 Report
Comments and Suggestions for Authors
In general this is an interesting study, well-motivated, with some findings of interest. This should be of interest to the readers.
The reviewer has the following suggests/comments/questions:
1. Table 1--For the animals need to include sex, age in table 1
2. Also Table 1--How was "central adiposity" measured?
3. In the discussion, please consider, with regards to the doses of the agents delivered, how does this compare to the range of human supplementation, also taking into account safety/toxicity?
4. Also for the discussion, please discuss the animal model used (dietary model of obesity with high sugar), and its applicability to humans--consider the amount/duration. This is very important with regards to all of the results. Different models could lead to different findings.
5. line 106 --"the C group" should be spelled out
6. In fig 4D, the treated MetS group have a pronounced elevation compared to all other groups, including untreated MetS, and interestingly, the level of Smad7 for that group was very similar to both treated and untreated controls. Why might this be the case?
Comments on the Quality of English Language
The reviewer suggests replacement of some terms ("reverts" to "reverses" for example) and some attention to verb tense in places. Overall this is well-written
Author Response
In general this is an interesting study, well-motivated, with some findings of interest. This should be of interest to the readers. The reviewer has the following suggests/comments/questions:
- Table 1--For the animals need to include sex, age in table 1
R= Thank you for your observation. We added the data in the description of table 1.
- Also Table 1--How was "central adiposity" measured?
R= Thank you for your observation. The intra-abdominal white adipose tissue was carefully dissected with scissors after euthanasia, wet weight was determined, and then the tissue was discarded. We have included this information in the methods section (4.3).
- In the discussion, please consider, with regards to the doses of the agents delivered, how does this compare to the range of human supplementation, also taking into account safety/toxicity?
R= Thank you for your suggestion. We have included more information regarding the safety of the polyphenols administered and the references.
- Also for the discussion, please discuss the animal model used (dietary model of obesity with high sugar), and its applicability to humans--consider the amount/duration. This is very important with regards to all of the results. Different models could lead to different findings.
R= The reviewer was right in her/his observation. The choice of a particular model requires the careful analysis of the variables or phenomenon to be studied, as multiple animal models of MetS are currently available (It can be induced by a high-fat, high-carbohydrate diet, or by using genetic models such as obese Zucker rats). Our MetS rat model used in this work, is easy to care for, relatively low cost and emulate specific lifestyle factors that humans exhibit in the pathophysiology of MetS. We have included this information in discussion section.
- line 106 --"the C group" should be spelled out
R= Corrected, thank you.
- In fig 4D, the treated MetS group have a pronounced elevation compared to all other groups, including untreated MetS, and interestingly, the level of Smad7 for that group was very similar to both treated and untreated controls. Why might this be the case?
R= The reviewer was right in her/his observation. We don’t know the exact reason for this effect due some studies in genetic Smad7−/− mice have been reported that the loss of muscular mass is associated with a decrease of Smad7 expression. Moreover, Smad7 is transcriptionally regulated by a variety of factors such as: TGF-β, activin, MAP kinases, and by transcription factors like AP-1; components that we have not yet analyzed in the muscles from our MetS rat model and might be altered.

Reviewer 2 Report
Comments and Suggestions for Authors
Abstract
Lines 25-26: remove capitalization on Resveratrol and Quercetin
Line 27: “In this work, we show that animals with MetS present” – describe the “animals” as male Wistar rats.
Line 28: you need to introduce the rat model and the experimental treatment timeline in the abstract. Explain the dosage and length of time resveratrol or quercetin was administered to the rats.
Lines 29-31: if any of these results mentioned in the abstract are statistically significant, you should include the P-value.
Introduction
Line 68: this is the first use of the abbreviation for TGF-β, so it should be described here as transforming growth factor-beta (TGF-β). Of note, the beta symbol is not appearing correctly in the manuscript.
Results
Line 109: first observation, “Control” should not be capitalized. This was also noted at lines 113, 123, 176, 209, 212, 225, 238, and other areas of the paper.
Line 140: provide the p-value in the text here for the significant increase in expression levels.
Lines 177-178: provide the p-value in the text here for the significant decrease in expression.
Line 205: Insulin resistance was previously abbreviated, so this term should be replaced with IR for consistency.
Discussion
Line 266: It is best to avoid referencing tables and figures in the discussion section since they have been referred to in the results section. Rephrase this sentence so that is does not direct to a table.
Line 267: change “animals” to male Wistar rats.
Line 271: briefly describe the animal models of obesity and MetS (rodents? provide strain and sex from the referenced studies) as it provides some relevance for making connections to your study and expansion to human studies.
Line 273: remove reference to Table 1.
Line 279: change “animals” to rats or male Wistar rats.
Lines 281-282: describe what animal study you are referring to here (rodents? provide strain)
Line 286: remove reference to figure 2.
Line 296: remove reference to figure 3.
Line 300: provide strain of obese mice and diabetic rats you are referring to in this sentence.
Line 308: the abbreviation TGF-β should have been defined at its first use at line 68. Of note, the symbol for beta is not formatted correctly.
Line 311: remove reference to figure 4.
Line 319: provide examples of some of the cells and tissues here in this sentence.
Line 330: remove reference to figure 6B
Line 331: provide the name of the strain of HDF mice
Lines 348-349: remove references to figure 7.
Methods
Line 355: provide the commercial source of the rats (breeder), or indicate if your institution runs a breeding colony of Wistar rats.
Lines 354-363: provide the name of the exact rat caging system used (model and manufacturer), and air handling details, provide the name and source of any bedding used (corn cob, wood chip, wire flooring, etc.), explain if any environmental enrichment was provided in the cages (gnawing sticks, covered shelters, etc), explain if the rats were housed in pairs or individually (or in groups).
Line 365: italicize ad libitum.
Line 369: change “rats were killed by decapitation” to rats were humanely euthanized.
Line 369: please include whether anaesthesia was included or not, and if it was, provide the name of the agent used.
Lines 369-370: describe how serum was processed, including the length of time blood was allowed to clot, any specialized tubes, and the centrifugation time/speed.
Line 374-375: Was serum insulin measured by ELISA? Please provide the model/manufacturer of the pate reader or instrument used to measure the assay.
Line 380: was the HDL measured in plasma as written, or should this be serum? If it is in fact plasma, please include the anticoagulant used, how long the blood rested prior to processing, and processing details.
Line 389: provide the exact name of the protease inhibitor cocktail used, and provide the supplier information
Line 493: the supplier details need to be updated for GraphPad: GraphPad Prism software (v5.03, Dotmatics, Boston, MA, USA). You should also mention in this section that you created your figures using GraphPad Prism in addition to the statistical analyses.
Conclusions
This is a great conclusion section; however, it should be mentioned around lines 500-504 that this work was completed in a rat model of SO and MetS.
Author Response
Abstract
Lines 25-26: remove capitalization on Resveratrol and Quercetin
R= Corrected, thank you.
Line 27: “In this work, we show that animals with MetS present” – describe the “animals” as male Wistar rats.
R= Corrected, thank you.
Line 28: you need to introduce the rat model and the experimental treatment timeline in the abstract. Explain the dosage and length of time resveratrol or quercetin was administered to the rats.
R= Thank you for your suggestion. We have included the information.
Lines 29-31: if any of these results mentioned in the abstract are statistically significant, you should include the P-value.
R= Thank you for your suggestion. We have included de p-values in the text.
Introduction
Line 68: this is the first use of the abbreviation for TGF-β, so it should be described here as transforming growth factor-beta (TGF-β). Of note, the beta symbol is not appearing correctly in the manuscript.
R= Thank you for your observation. We correctly placed the abbreviation and corrected the symbol.
Results
Line 109: first observation, “Control” should not be capitalized. This was also noted at lines 113, 123, 176, 209, 212, 225, 238, and other areas of the paper.
R= The word was corrected throughout the manuscript, thank you.
Line 140: provide the p-value in the text here for the significant increase in expression levels.
R= Thank you for your suggestion. We have included de p-value in the text.
Lines 177-178: provide the p-value in the text here for the significant decrease in expression.
R= Thank you for your suggestion. We have included de p-value in the text.
Line 205: Insulin resistance was previously abbreviated, so this term should be replaced with IR for consistency.
R= Corrected, thank you.
Discussion
Line 266: It is best to avoid referencing tables and figures in the discussion section since they have been referred to in the results section. Rephrase this sentence so that is does not direct to a table.
R= Corrected, thank you.
Line 267: change “animals” to male Wistar rats.
R= Corrected, thank you.
Line 271: briefly describe the animal models of obesity and MetS (rodents? provide strain and sex from the referenced studies) as it provides some relevance for making connections to your study and expansion to human studies.
R= Thank you for your suggestion. We have included the information of the rodent models.
Line 273: remove reference to Table 1.
R= Thank you for your suggestion. We have deleted the reference.
Line 279: change “animals” to rats or male Wistar rats.
R= Corrected, thank you.
Lines 281-282: describe what animal study you are referring to here (rodents? provide strain)
R= Thank you for your suggestion. We have included the information of the rodents.
Line 286: remove reference to figure 2.
R= Thank you for your suggestion. We have deleted the reference.
Line 296: remove reference to figure 3.
R= Thank you for your suggestion. We have deleted the reference.
Line 300: provide strain of obese mice and diabetic rats you are referring to in this sentence.
R= Thank you for your suggestion. We have included the information of the rodents to clarify.
Line 308: the abbreviation TGF-β should have been defined at its first use at line 68. Of note, the symbol for beta is not formatted correctly.
R= Corrected, thank you.
Line 311: remove reference to figure 4.
R= Thank you for your suggestion. We have deleted the reference.
Line 319: provide examples of some of the cells and tissues here in this sentence.
R= Thank you for your suggestion. We have included some examples.
Line 330: remove reference to figure 6B
R= Thank you for your suggestion. We have deleted the reference.
Line 331: provide the name of the strain of HDF mice
R= Thank you for your suggestion. We have included the strain of mice.
Lines 348-349: remove references to figure 7.
R= Thank you for your suggestion. We have deleted the references.
Methods
Line 355: provide the commercial source of the rats (breeder), or indicate if your institution runs a breeding colony of Wistar rats.
R= The Wistar male rats employed in this study were provided by the Bioterio of our Institution. We included this information in 4.1 section.
Lines 354-363: provide the name of the exact rat caging system used (model and manufacturer), and air handling details, provide the name and source of any bedding used (corn cob, wood chip, wire flooring, etc.), explain if any environmental enrichment was provided in the cages (gnawing sticks, covered shelters, etc), explain if the rats were housed in pairs or individually (or in groups).
R= We included the details in 4.2 section from methods.
Line 365: italicize ad libitum.
R= Corrected, thank you.
Line 369: change “rats were killed by decapitation” to rats were humanely euthanized.
R= Corrected, thank you.
Line 369: please include whether anesthesia was included or not, and if it was, provide the name of the agent used.
R=We did not use anesthesia prior to euthanasia. We added this information in methods section.
Lines 369-370: describe how serum was processed, including the length of time blood was allowed to clot, any specialized tubes, and the centrifugation time/speed.
R= Thank you for your suggestion. We have detailed the method for obtaining the serum in methods section (4.2).
Line 374-375: Was serum insulin measured by ELISA? Please provide the model/manufacturer of the pate reader or instrument used to measure the assay.
R= Serum insulin concentration was measured by radioimmunoassay (RIA) previously developed by our working group and carefully validated. Our method is a competitive radioimmunoassay in liquid phase with precipitation using a second antibody, and a polyclonal antibody produced in guinea pigs against recombinant rat insulin. Insulin was measured using a gamma counter Cobra II (Packard Instrument Company, Inc., Connecticut, USA). We included the information of the counter in methods section.
Line 380: was the HDL measured in plasma as written, or should this be serum? If it is in fact plasma, please include the anticoagulant used, how long the blood rested prior to processing, and processing details.
R= Thank you for your observation. We determined the HDL content in serum, so we corrected the term.
Line 389: provide the exact name of the protease inhibitor cocktail used, and provide the supplier information
R= Thank you for your observation. We added the supplier information.
Line 493: the supplier details need to be updated for GraphPad: GraphPad Prism software (v5.03, Dotmatics, Boston, MA, USA). You should also mention in this section that you created your figures using GraphPad Prism in addition to the statistical analyses.
R= Thank you for your suggestion. We have included the supplier details.
Conclusions
This is a great conclusion section; however, it should be mentioned around lines 500-504 that this work was completed in a rat model of SO and MetS.
R= Thank you for your suggestion. We have included the information.
Comments on the Quality of English Language
The reviewer suggests replacement of some terms ("reverts" to "reverses" for example) and some attention to verb tense in places. Overall this is well-written
R= Minor grammatical errors in the manuscript were corrected and we have reworked some sentences to avoid duplication (iThenticate report).
